# Quantum Monte Carlo detection of SU(2) symmetry breaking in the participation entropies of line subsystems

## David J. Luitz[1] and Nicolas Laflorencie[2*]

**1** Department of Physics and Institute for Condensed Matter Theory, University of Illinois at Urbana-Champaign, Urbana, Illinois 61801, USA
**2** Laboratoire de Physique Théorique, IRSAMC, Université de Toulouse, CNRS, 31062 Toulouse, France

\* laflo@irsamc.ups-tlse.fr

## Abstract

Using quantum Monte Carlo simulations, we compute the participation (Shannon-Rényi) entropies for groundstate wave functions of Heisenberg antiferromagnets for one-dimensional (line) subsystems of length $L$ embedded in two-dimensional ($L \times L$) square lattices. We also study the line entropy at finite temperature, *i.e.* of the diagonal elements of the density matrix, for three-dimensional ($L \times L \times L$) cubic lattices. The breaking of SU(2) symmetry is clearly captured by a universal logarithmic scaling term $l_q \ln L$ in the Rényi entropies, in good agreement with the recent field-theory results of Misguish, Pasquier and Oshikawa [26]. We also study the dependence of the log prefactor $l_q$ on the Rényi index $q$ for which a transition is detected at $q_c \simeq 1$.



# 1   Introduction

The entanglement of ground states in quantum many body systems has been found to reflect fundamental features and universal aspects [1–3], such as spontaneous symmetry breaking, topological properties, as well as geometrical aspects of the entanglement bipartition (*e.g.* corner contributions). In the case of a system that spontaneously breaks a continuous symmetry, recent analytical [4–9] and numerical [7, 10–13] results indicate that subleading corrections to the scaling of the entanglement entropy are logarithmic with system size, with a prefactor proportional to the number of Nambu-Goldstone modes $n_{NG}$ associated to the broken symmetry. Computationally, the entanglement entropy of a subsystem of a higher dimensional system remains hard to access, despite recent progress in quantum Monte Carlo methods [11, 14–16]. Similarly, DMRG calculations of ground-states in dimension higher than 1 are difficult due to the underlying area law for entanglement entropy [17].

Here, we resort to a somewhat simpler quantity that is known to capture universal information in its subleading terms beyond a generic multifractal volume-law scaling [18–23]: The (basis-dependent) *participation entropy*, defined for a quantum state $|\Phi\rangle$ in a given computational discrete basis $\{|i\rangle\}$ by

$$S_q^P = \frac{1}{1-q} \ln \sum_i |\langle\Phi|i\rangle|^{2q}, \tag{1}$$

where the sum is taken over all possible basis states whose number can grow exponentially with the number of sites of the many-body system. $S_q^P$ can be efficiently calculated in QMC calculations [23–25] *in the computational basis* thanks to the replica trick and, even simpler, in the case of the infinite Rényi index by recording the probability of the most probable basis states, which need not be unique. By looking only at the diagonal elements of the (reduced) density matrix, one can numercially study considerably larger systems compared to calculations of the basis independent entanglement entropy, thus allowing to explore universal features in the finite-size scaling properties of $S_q^P$.

The participation entropy can be computed either for a full system or for subsystems after performing a bipartition. Typically, it displays a *volume-law* scaling in the size of the system (or the subsystem) and therefore grows faster than an area-law. Since the computational efficiency depends directly on the value of the participation entropy, which boils down to an efficient estimation of very small probabilities [16], it is important to choose the subsystem such that it has the *minimal volume* while still capturing the relevant long wave-length physics. Also to avoid geometrical (e.g. corner) contributions, it is useful to consider subsystems with smooth boundaries. In this work, we study the minimal subsystem with this property, namely a one-dimensional line, embedded in periodic square and cubic lattices (depicted in Fig. 1). For SU(2) symmetry breaking, we show that the logarithmic correction in the entanglement entropy, which reflects the number of Nambu-Goldstone modes $n_{NG} = 2$, also appears in the participation entropy, and that this fundamental feature is captured by the minimal line sub-

system considered. Our QMC results are found in good agreement with the recent field-theory calculations of Misguich, Pasquier and Oshikawa (MPO) [26].

This paper is structured as follows: In Section 2, we present the lattice spin models, briefly discuss the analytical results from MPO [26], and present the numerical method. In Section 3, our quantum Monte Carlo results for square and cubic lattices are presented for both finite and infinite Rényi indices $q$, for which clear additive logarithmic corrections are found, in agreement with MPO [26]. Furthermore, a transition in function of the Rényi parameter $q$ is detected for $q_c \simeq 1$, thus suggesting a possible true thermodynamic transition for the corresponding one dimensional entanglement Hamiltonian. Finally in Section 4 we discuss our results, and give a few possible future directions.

## 2 Models and methods

### 2.1 Quantum spin models

We consider two $S = 1/2$ quantum antiferromagnets in two and three dimensions. The first Hamiltonian is the so-called $J_1 - J_2$ model, defined on a square lattice by

$$\mathcal{H} = J_1 \sum_{\langle ij \rangle} \vec{S}_i \cdot \vec{S}_j + J_2 \sum_{\langle\langle ij \rangle\rangle} \vec{S}_i \cdot \vec{S}_j, \tag{2}$$

where $\vec{S}$ are spin-1/2 operators, interactions act between nearest neighbours (n.n.) $\langle ij \rangle$ and next nearest neighbours (n.n.n.) $\langle\langle ij \rangle\rangle$ along the diagonals of the square lattice (see panel (a) of Fig. 1). For this work, we will consider antiferromagnetic n.n. interactions $J_1 > 0$ and *ferromagnetic* n.n.n. interactions $J_2 < 0$, for which it is known that antiferromagnetic long-range order exists in the ground-state. In the thermodynamic limit, the SU(2) symmetry is therefore expected to be broken in the ground-state of $\mathcal{H}$, with two associated Nambu-Goldstone modes.

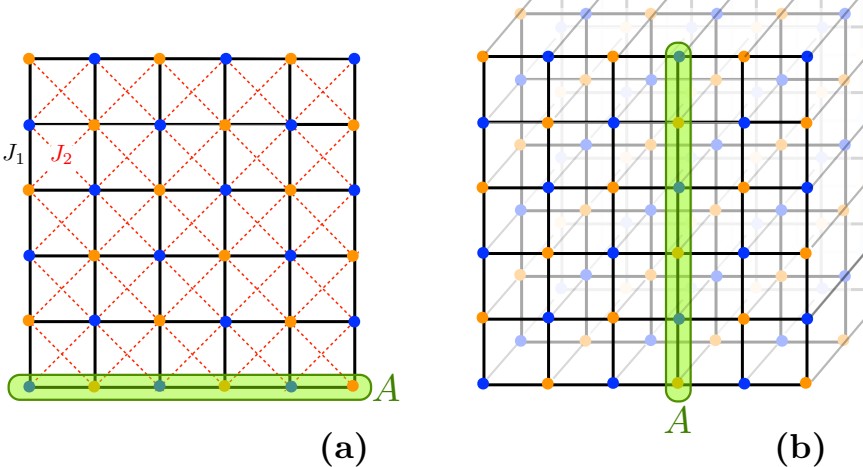

Figure 1: Schematic picture for the two lattice antiferromagnets. (a) The $J_1 - J_2$ Hamiltonian defined on a two-dimensional square lattice Eq. (2). (b) The nearest-neighbor Heisenberg model Eq. (3) on a three-dimensional cubic lattice. In both cases, the line shaped subsystem $A$ is shown in green.

The interest of adding the n.n.n. interaction $J_2$ is two-fold: first, we would like to check the universality of our results with respect to the number of Nambu-Goldstone modes (which are

two, independent of $J_2 < 0$) by considering several values of $J_2$. Also as $|J_2|$ is increased, the antiferromagnetic long-range order gets stronger (i.e. larger values of the order parameter), which will in turn translate in lower participation entropies in the $\{S^z\}$ basis and facilitate our numerical calculations by increasing the associated probabilities that have to be sampled in the Monte Carlo simulation.

The second spin model, defined on the three-dimensional cubic lattice (see panel (b) of Fig. 1), is the standard n.n. Heisenberg $S = 1/2$ antiferromagnet

$$\mathcal{H} = J \sum_{\langle ij \rangle} \vec{S}_i \cdot \vec{S}_j, \tag{3}$$

which has Néel long-range order at finite temperature for $T/J < T_c$ with $T_c = 0.94408(2)$ [24].

## 2.2 Analytical predictions

In Ref. [26], MPO have studied the finite size scaling of the Shannon-Rényi participation entropies Eq. (1) for groundstates of long-range ordered systems with a broken continuous symmetry using a combination of free-field theory to treat the oscillator modes (the spin waves), supplemented by phase space arguments to account for the spin rotation symmetry in finite space, yielding

$$S_{q>1}^{\mathrm{P}}(N) = \mathcal{O}(N) + \frac{n_{\mathrm{NG}}}{4} \frac{q}{q-1} \ln N + \text{subleading terms}, \tag{4}$$

where $N$ is the number of sites and $n_G$ the number of Nambu-Goldstone bosons associated with the symmetry breaking. For $q = 1$, only the oscillators contribute to the logarithmic correction, predicted to be $-\frac{n_{\mathrm{NG}}}{4} \ln N$ [26].

As recalled in Ref. [26], logarithmic corrections of the form Eq. (4) have been previously observed in a numerical work [23] using large scale quantum Monte Carlo simulations for the entropies computed at $q = 2, 3, 4, \infty$, in groundstate wave functions of full two-dimensional systems. In both U(1) and SU(2) cases, there is a reasonably good agreement between numerics and the prediction Eq. (4). Nevertheless, the obtention of a correct estimate of the prefactor in front of the log turned out to be quite difficult, simply due to the very slow growth of a logarithmic term with $N$, which despite substantial numerical efforts [23] was limited to system sizes ranging from $4 \times 4$ up to $20 \times 20$. This delicate issue is also present for the numerical study of finite size log corrections in the entanglement entropy [7, 10–13].

## 2.3 Line subsystem

Interestingly, as first discussed in a series of papers [24, 25, 27, 28], one can spatially restrict the computation of the entropies to a finite subsystem, instead of the entire lattice, by tracing out degrees of freedom in the complement of the subsystem and considering the *diagonal part* of the obtained reduced density matrix. A very useful bipartition consists in taking a periodic line of length $L$ in a square $L \times L$ or cubic $L \times L \times L$ lattice, thus allowing to reach much larger linear system sizes up to $L = 64$. This was done for instance in Ref. [24] where an additive constant term $b_\infty^* \simeq 0.41$ was found for 3d Heisenberg critical points, and conjectured to be universal for 3d O(3) criticality.

In Ref. [26], MPO have also studied such a case with a one-dimensional subsystem of length $L$ embedded in an $L \times L$ torus, and they found similar logarithmic corrections for continuous symmetry breaking states. More precisely, the following scaling is expected

$$S_{\mathrm{line}, q>1}^{\mathrm{P}}(L) = \mathcal{O}(L) + \frac{n_{\mathrm{NG}}}{2} \frac{q}{q-1} \ln L + \text{subleading terms}, \tag{5}$$

with again the number of Nambu-Goldstone modes $n_{\text{NG}}$ controlling the logarithmic correction. In the case $q = 1$, the logarithmic term is expected to vanish[1], and there is currently no analytical prediction for the case of $q < 1$. In the rest of this paper we will aim to verify this prediction for the line participation entropy using large scale quantum Monte Carlo simulations.

## 2.4 Quantum Monte Carlo

We perform extensive quantum Monte Carlo simulations using the stochastic series expansion (SSE) algorithm together with our recently introduced [16] improved estimator for the participation entropies

$$S_{q,\,\text{line}}^{\text{P}} = \frac{1}{1-q} \ln \sum_i \left( \rho_{ii}^{\text{line}} \right)^q \tag{6}$$

by virtue of the replica trick. Here, $\rho^{\text{line}} = \frac{1}{Z} \text{Tr}_{\bar{A}} e^{-\beta \mathcal{H}}$ is the reduced density matrix of the line shaped subsystem $A$ (*cf.* Fig. 1) and the participation entropy only depends on its diagonal elements $\rho_{ii}^{\text{line}}$ in the local spin basis.

Note that the system is translation invariant and we exploit translation symmetry along the subsystem as well as different translations of the subsystem in addition to the spin flip symmetry to enhance the quality of our estimator for the participation entropy. Here, we only focus on line subsystems and therefore skip the subscript "line".

# 3 Quantum Monte Carlo results

## 3.1 Most probable state and $S_\infty^{\text{P}}$

### 3.1.1 Two dimensions

The limit of $q \to \infty$ is easier to treat in QMC calculations, as it can be obtained by counting the occurrence of the most probable basis state (which are the two Néel states in this case [23,25]):

$$S_\infty^{P} = -\ln \left( \max_i \rho_{ii}^{\text{line}} \right). \tag{7}$$

As the line shaped subsystem can be realized in $2L$ different ways due to translation and rotation (by $\frac{\pi}{2}$) invariance of the Hamiltonian, we exploit these possibilities in order to improve the statistics.

Our results are displayed in Fig. 2 for various values of the n.n.n. coupling $J_2$. Here, we study the system at zero temperature and sample directly the pure groundstate wavefunction, which we obtain by performing simulations at low temperatures such that the participation entropy is converged to the groundstate results. We find that inverse temperatures $\beta J_1 = 4L$ are sufficiently low for this purpose.

The logarithmic scaling with system size is visible in the curvature of the participation entropy as a function of system size and we estimate the prefactor $l_\infty$ by performing fits of the form

$$S_\infty^{\text{P}} = a_\infty L + l_\infty \ln L + b_\infty + c_\infty / L. \tag{8}$$

Note that due to the higher precision of our results for $S_\infty^{\text{P}}$ compared to finite values of $q$, fits including the correction term $c_\infty / L$ are stable and yield slightly better fit qualities than without this term. We systematically reduce the fit range $[L_{\text{min}}, 128]$ by increasing $L_{\text{min}}$, thus excluding smaller system sizes and study the evolution of the logarithmic term as a function of $L_{\text{min}}$. While the stability of the fit clearly decreases for smaller fit ranges, finite size effects

---

[1]G. Misguich, private communication.

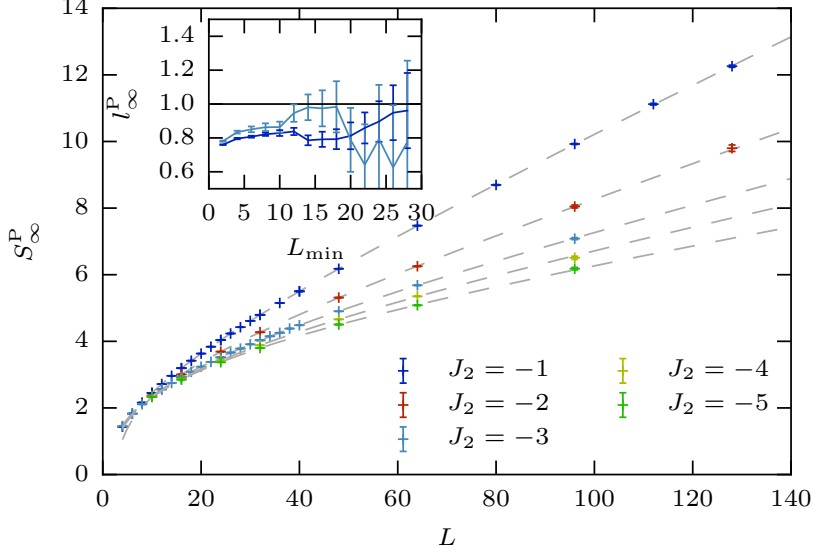

Figure 2: Participation entropy $S_\infty^P$ of the line shaped subsystem for the $T = 0$ two-dimensional (panel (a) of Fig. 1) $S = 1/2$ Heisenberg model [Eq. (2)] at different values of $J_2$. The inset shows the prefactor of the logarithmic term obtained from fits to the form Eq. (8), using different fit windows $[L_{\min}, L_{\max}]$ as a function of $L_{\min}$ for the two values of $J_2 = -1$ and $J_2 = -3$, for which we have collected data on the finest grid in system size. $L_{\max}$ is the maximal linear system size, ranging from 96 to 128. Our results are consistent with a universal logarithmic term with $l_\infty = n_{NG}/2 = 1$, since for fit windows that include only large system sizes, the obtained $l_\infty$ is close to 1 within errorbars.

are expected to be reduced systematically and indeed our result seems to move towards $l_\infty$ before large errorbars make the fit unreliable.

The result of our fit analysis shown in the inset of Fig. 2 suggests that, as $L_{\min}$ is increased, the coefficient of the logarithmic subleading term tends to a constant value, close to $l_\infty = 1$. Once again, this result appears to be independent of $J_2$, thus confirming the universal character.

### 3.1.2 Three dimensions

Let us now move on to more general mixed states. We simulate a three dimensional Heisenberg antiferromagnet on a cubic lattice at finite temperature $T$ (Eq. (3) with $J$ set $= 1$). This system is known to exhibit a thermodynamic phase transition from a paramagnet to an antiferromagnet at $T_c = 0.94408(2)$ [24] breaking the continuous $SU(2)$ symmetry below $T_c$ with $n_{NG} = 2$ Nambu-Goldstone bosons. We will again use a unidimensional subsystem $A =$ line as depicted in Fig. 1 (b) and entirely trace out the rest of the system $(\overline{\text{line}})$ to obtain the reduced density matrix $\rho^{\text{line}} = \frac{1}{Z} \text{Tr}_{(\overline{\text{line}})} e^{-\beta \hat{H}}$ where $\beta = 1/T$ is the inverse physical temperature.

Just as in the case of groundstate simulations, we are able to obtain only the diagonal matrix elements of $\rho^{\text{line}}$ *in the computational basis* in principle, but will restrict our study here to the Rényi index $q = \infty$, amounting to the calculation of the maximal diagonal element, which in the case of this model is always the Néel state on the subsystem, no matter in which phase the system is. From this matrix element, the participation entropy $S_\infty^P$ is obtained for different system sizes (up to $L = 80$) and temperatures by Monte Carlo sampling of the thermal density matrix.

The results for $S_\infty^P$ are depicted in Fig. 3 for three representative temperatures. (i) In

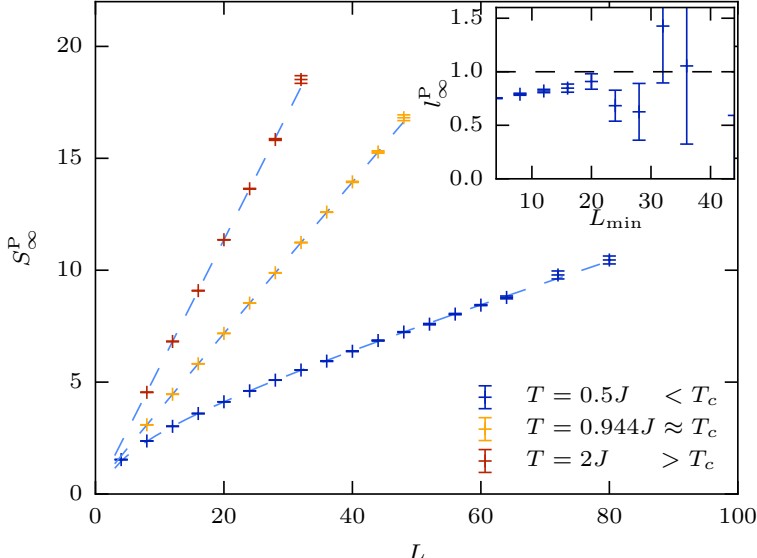

Figure 3: Participation entropy $S_\infty^{\mathrm{P}}$ of the line shaped subsystem in a three dimensional antiferromagnet (panel (b) of Fig. 1) at different temperatures. Below the critical temperature $T_c$, the thermodynamic limit ordering with a breaking of the continuous $SU(2)$ symmetry shows up on finite systems as the participation entropy acquires a logarithmic correction term. The inset shows the prefactor of this logarithmic term obtained from fits to the form $S_\infty^{\mathrm{P}} = a_\infty L + l_\infty \ln L + b_\infty$, which agrees well with $l_\infty = 1$.

the disordered paramagnetic phase at $T = 2 > T_c$ where $S_\infty^{\mathrm{P}} = a_\infty L$ [25]. (ii) At criticality $T = 0.944 \approx T_c$, where $S_\infty^{\mathrm{P}} = a_\infty L + b_\infty$ with $b_\infty \simeq 0.41$ [25]. (iii) In the Néel ordered regime at $T = 0.5 < T_c$ where the ordering is qualitatively signaled by a strong reduction of the overall participation entropy, which means that the weight of the Néel state drastically increases as the temperature is reduced below $T_c$. More quantitatively, the entropy as a function of system size acquires a curvature, which reflects the emergence of a logarithmic term. We perform the same analysis of the prefactor of the logarithmic term and observe that as the fit window moves to larger system sizes, this term approaches $l_\infty = 1$. Note that if $L_{\min}$ is too large, the fit becomes unstable and is no longer reliable, as the domain to observe the logarithm is too small and the fit can not "see" the curvature of the function.

## 3.2 Results at finite Rényi index $q$ for two dimensions

### 3.2.1 Replica trick results for integer Rényi index

Let us now consider participation entropies of the line shaped subsystem embedded in a two-dimensional square lattice (model Eq. (2), panel (a) of Fig. 1) *at finite Rényi indices $q = 2, 3, 4$* obtained from simulations using 4 replicas. We have performed groundstate calculations for system sizes from $L = 4$ to $L = 40$ in order to extract the scaling with system size for different values of $q$.

Our QMC results for participation entropies of various Rényi indices $q$ are displayed in Fig. 4 which clearly show that the subsystem participation entropy grows with system size with a logarithmic correction that leads to a visible curvature. In an attempt to evaluate the subleading logarithmic scaling term, we perform fits of the form

$$S_q^{\mathrm{P}} = a_q L + l_q \ln L + b_q \tag{9}$$

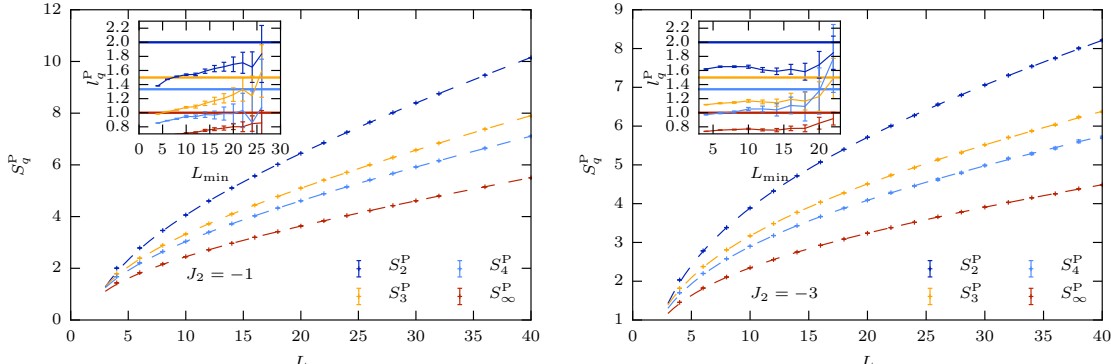

Figure 4: Participation entropies $S_q^{\mathrm{P}}$ of the line shaped subsystem $A$ for $J_2 = -1$ (left) and $J_2 = -3$ (right) as a function of the linear system size $L$. The dashed lines display our best fits over the range $L \in [10, 40]$. The logarithmic term is visible and the inset shows the result of a fit to the form Eq. (9) over the fit range $[L_{\min}, 40]$ as a function of $L_{\min}$. Horizontal lines are guides to the eye for $q/(q-1)$ for $q = 2, 3, 4$ and $q = \infty$.

systematically over different fit ranges $[L_{\min}, 40]$, where $L = 40$ is our largest linear system size available. Our results are consistent with the form

$$l_q = \frac{q}{q-1}, \tag{10}$$

indicated by horizontal lines (for $q = 2, 3, 4$ and $q = \infty$) in the insets of Fig. 4 (left and right). It is however obvious that the available range of system sizes does not allow for a more conclusive result. We have also tried to include additional terms for the scaling with system size, however the quality of our data does not yield stable fits in this case.

Quite importantly, the result Eq. (10) appears to hold independent of the value of $J_2$ (other values of $J_2$ not shown), suggesting its universal nature. Below we further explore the $q$-dependence for any (non-integer) value of the Rényi parameter.

### 3.2.2   Direct calculation $\forall q$ and Rényi index transition

Instead of using the replica trick as above, which is restricted to small integer values of $q$, we can directly compute the histogram of the $2^L$ basis states. Translation invariance of the line subsystem allows to deal with lengths up to $L = 30$, which corresponds to more than $10^9$ basis states. We note that the drawback of using the histogram method is that it can not exploit the exponential improvement of statistics obtained in our improved replica estimator [16], however it is currently the only possibility to study fractional Rényi indices as well as $q < 2$.

We focus on the ground state of the two-dimensional ($L \times L$) antiferromagnetic Heisenberg model Eq. (2) at $J_2 = -J_1$ for which we built the histogram of the line subsystem during SSE simulations performed at inverse temperature $\beta J_1 = 4L$ (which corresponds to sufficiently low temperature such that all results are converged to the ground state). Results for the participation entropies are shown in panel (a) of Fig. 5 for some representative values of $q$. In order to extract the prefactor of the logarithmic correction we perform fits to the following form

$$S_q^{\mathrm{P}}(L) = a_q L + l_q \ln L + b_q + c_q/L \tag{11}$$

for four different sliding windows containing 11 points ranging from $L_{\min} = 4, 6, 8, 10$ to $L_{\max} = 24, 26, 28, 30$. For all results in this paper we have checked that the inclusion of additional subleading corrections in the form of powers of $1/L$ do not change significantly the

results and we vary the form of the fit function on a case by case basis depending on which form gives the best fit. Since these subleading terms are only of relevance for the smallest system sizes, the particular choice is not important.

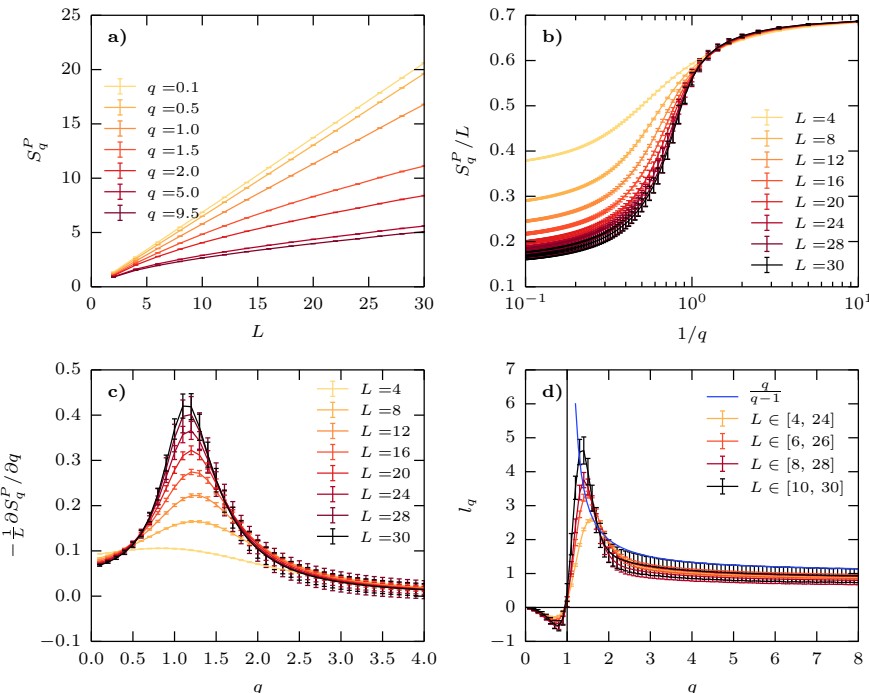

Figure 5: QMC results for the groundstate participation entropies $S_q^{\mathrm{P}}$ of a periodic line embedded in the $L \times L$ two-dimensional $J_1 - J_2$ Hamiltonian Eq. (2) at $J_1 = 1$, $J_2 = -1$, computed directly from the histogram of sampled basis states. (a) Entropies as a function of the linear system size $L$ for some representative values of the Rényi parameter $q$. (b) Intensive part $S_q^{\mathrm{P}}/L$ plotted against $1/q$ for varying lengths $L$. An inflection point is demonstrated in panel (c) where the derivative is peaked around $q_c \simeq 1$. (d) $q$-dependence of the prefactor $l_q$ of the logarithmic correction obtained from fits to the form Eq. (11) performed over various windows, as indicated on the plot. The analytic prediction Eq. (10) is also shown for $q > 1$ (blue line). Panels (c-d) clearly show a Rényi index transition at $q_c \simeq 1$.

Results for the log prefactor $l_q$ are plotted in panel (d) of Fig. 5 where a non-trivial $q$-dependence is clearly visible. First, for $q > 1$ our estimate compares quite well with the analytical prediction in Eq. (10), and this tendency becomes better for increasing system size. When $q \to 1$, the diverging behavior of $l_q$ is quite well reproduced. Interestingly, there is a qualitative change at $q_c \simeq 1$ where the logarithmic term is discontinuous and changes sign, with a value at the transition compatible with $l_{q_c} = 0$. One can also detect this Rényi transition in panel (b-c) of Fig. 5 where $S_q/L$ plotted against $1/q$ displays an inflection point at the transition which clearly appears in the derivative $\partial S_q^{\mathrm{P}}/\partial q$ as a maximum at $q_c \simeq 1$, apparently diverging with the system size $L$.

Rényi index transitions, *i.e.* transitions observed in some universal corrections to the participation entropies as a function of the Rényi index $q$, have already been discussed for the participation entropies of full systems for Luttinger liquids [18, 20, 21] and for the quantum Ising model [19, 23, 28], but to the best of our knowledge never for susbsystems. This is therefore the first example of such a transition, which may be interpreted as a finite temperature transition occuring in the so-called entanglement Hamiltonian. Indeed, the Rényi parameter $q$ plays the role of an effective inverse temperature in a description based on the entanglement

Hamitonian $\mathcal{H}_E$, defined on the line subsystem by $\rho^{\text{line}} = \exp(-\mathcal{H}_E)$, where $\rho^{\text{line}}$ is the reduced density matrix. A thermal phase transition for $\mathcal{H}_E$ would be easily detected in the groundstate *entanglement* entropy $S_q^E = (1-q)^{-1} \ln\left[\text{Tr}\left(\rho^{\text{line}}\right)^q\right]$ which, within such a formalism, is directly related to the free energy associated to $\mathcal{H}_E$ at temperature $T = 1/q$ by $F = (1-1/q)S_q^E$. Therefore, a singularity in $F$ at $T_c$ would directly show up in $S_{q_c}^E$ at $q_c = 1/T_c$. Here, one may suspect that the singularity observed in Fig. 5, not for $S_q^E$ but for $S_q^P$ at $q_c \simeq 1$ is an indirect evidence for a thermodynamic transition in $\mathcal{H}_E$. In Ref. [25] it was conjectured for the same one dimensional bipartition that the entanglement Hamiltonian should be long-ranged, with power-law decaying unfrustrated pairwise couplings, which would be consistent with an $\mathcal{O}(1)$ value for $T_c$.

## 4 Conclusions

Using large-scale quantum Monte Carlo simulations, we find that subleading terms (beyond volume law) of participation entropies for a line-shaped subsystem in Néel ordered Heisenberg antiferromagnets scale logarithmically $l_q \log L$, with a universal coefficient $l_q$, proportional to the number of Nambu-Goldstone modes, thus confirming analytical predictions by Misguich, Pasquier and Oshikawa [26]. Furthermore our numerical data are in very good agreement with $l_q = \frac{q}{q-1}$ for $q > 1$, suggesting that the physics is completely dominated by the $q = \infty$ limit, *i.e.* the coefficient of the Néel state, with a corresponding subleading term coefficient $l_\infty = 1$. Remarkably, a Rényi index transition is observed at $q_c \simeq 1$, with both a disappearance of positive logarithmic corrections and a singularity in the derivative $\partial S_q^P / \partial q$. A better understanding of this phenomenon is needed, perhaps in terms of a thermal transition in the entanglement Hamiltonian.

## Acknowledgements

We thank Grégoire Misguich for insightful discussions and Fabien Alet for earlier collaborations on related topics.

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
