# Peer review of "Quantum Monte Carlo detection of SU(2) symmetry breaking in the participation entropies of line subsystems"

_SciPost Physics, doi:SciPost Phys. 2, 011 (2017)_

## Round 1 · Referee Report · Anonymous · 2017-1-30

Strengths

(1) Timely contribution to a fundamental property of quantum many-body systems in terms of entanglement.
(2) Useful contribution from a numerical method in good accord with recent field theory predictions.
(3) Clear presentation of the paper's overall strategy and methodology.
(4) Interesting observation of the transition in the q-dependece of the log-prefactor, calls for further work.

Weaknesses

(1) Only minor issues, to be detailed below.

Report

The authors use a quantum Monte Carlo approach to calculate the participation entropy in 2D (in the groundstate) and 3D (finite T) Heisenberg antiferromagnets for an embedded 1D line subsystem. They identify a logarithmic scaling term in accord with previous field-theory predicions, which quantifies the spontaneous symmetry-breaking contribution from Goldstone modes. Furthermore, they identify an interesting transition in the dependence of the log-prefactor on the Renyi index, which they associate with a possible thermal transition in the entanglement Hamiltonian.

The numerical results are sound and have been carefully analyzed, including an error analyis in the fitting procedure to the scaling laws. The employed analysis, focusing on the sublattice particpation is original and in good overall accord with recent field-theory predictions with respect to the Goldstone-mode contribution to the log-corrections, which is known to be difficult to extract in e.g. a direct analysis of the entanglement entropy. The obervation of a transition in the Renyi-index dependence and its linkage to the entanglement entropy points towards an interesting perspective for future analysis.

The paper is well written and explains very clearly the overall idea and strategy behind the taken approach. Nevertheless, I think that the presentation should be improved with respect to the points given below:

Requested changes

(1) In addition to Ref. 1, several other general reviews of the topic exist, which should also be cited.

(2) In addition to Ref. 2, the authors should cite also O. A. Castro-Alvaredo and B. Doyon, Phys. Rev. Lett. 108, 120401 (2012), with regards to theory work on the general Goldstone-mode contribution.

(3) At the beginning of 3.1.1, the actual estimator for the limit q->\infnty should be stated (similar to eq (6)) for clarity.

(4) An explicit algorithm should be cited after "Here, we study the system at zero temperatures..." in the second paragraph of 3.1.1.

(5) For the inset of Fig. 2, only two values of J2 are considered, while in the main panel of that figure, more values of J2 are shown. A reason should be given for this selection of J2 values for the inset (it appears that for these two values of J2, data on many more small systems was collected). Why is the most simple case of J2=0 not considered at all?

(6) Why is the convergence in the inset of Fig. 2 from below? Can this be understood?

(7) It appears more appropriate from the present analysis to write that "a transition is detected at q_c", instead of "phase transition", as the later has a defined meaning, and it is unclear, if this really applies here.

  • validity: top
  • significance: high
  • originality: high
  • clarity: high
  • formatting: perfect
  • grammar: excellent

Author:  David J. Luitz  on 2017-02-21

(in reply to Report 1 on 2017-01-30)
Category:
answer to question

We are grateful to the referee for the detailed comments on our manuscript and for the very positive evaluation. We reply to the points raised by the referee below.

(1) “In addition to Ref. 1, several other general reviews of the topic exist, which should also be cited.”

We have included the following references:
“Entanglement in many-body systems” by Luigi Amico, Rosario Fazio, Andreas Osterloh, and Vlatko Vedral, Rev. Mod. Phys. 80, 517 (2008) and
“Entanglement entropy and conformal field theory” by Pasquale Calabrese, John Cardy, J.Phys.A42:504005,2009

(2) “In addition to Ref. 2, the authors should cite also O. A. Castro-Alvaredo and B. Doyon, Phys. Rev. Lett. 108, 120401 (2012), with regards to theory work on the general Goldstone-mode contribution.”

We have added a citation to this article along with the Ref. to Metlitsky and Grover 2011.

(3) “At the beginning of 3.1.1, the actual estimator for the limit q->\infnty should be stated (similar to eq (6)) for clarity.”

We have included an explicit expression for the estimator for S_\infty.

(4) “An explicit algorithm should be cited after "Here, we study the system at zero temperatures..." in the second paragraph of 3.1.1.”

In the new version of the manuscript, we describe the procedure how we sample the groundstate using sufficiently low temperatures in a finite temperature SSE algorithm, such that our results are converged to the ground state values.

(5) “For the inset of Fig. 2, only two values of J2 are considered, while in the main panel of that figure, more values of J2 are shown. A reason should be given for this selection of J2 values for the inset (it appears that for these two values of J2, data on many more small systems was collected). Why is the most simple case of J2=0 not considered at all?”

The referee is right that we show the convergence of l_infty only for two examples (J2=-1 and J2=-3) to avoid clutter in the inset and because we collected data on a very fine grid of system sizes for these values of J2. While theoretically J2=0 is indeed the simplest case, the entropies tend to be much larger (since the J2 term reinforces the Néel order). The probabilities that have to be estimated in our QMC method are exponentially small in the entropy and for the case J2=0 it is therefore much more difficult to obtain results with sufficiently small errorbars.

(6) “Why is the convergence in the inset of Fig. 2 from below? Can this be understood?”

There is currently no good analytic understanding of subleading scaling terms of the participation entropies, after the logarithmic term. We suspect that one can write down an expansion in terms of powers of 1/L (first term included in (7)) and the detailed behavior of these subleading terms (which vanish for larger systems) could be responsible for the convergence of l_infty. For example, in a previous work http://journals.aps.org/prb/pdf/10.1103/PhysRevB.91.155145 we observed that the way the log prefactor converge towards its thermodynamic limit value can depend on the subleading terms included in the fitting form.

(7) “It appears more appropriate from the present analysis to write that "a transition is detected at q_c", instead of "phase transition", as the later has a defined meaning, and it is unclear, if this really applies here.”

We agree and have changed this statement as proposed by the referee, replacing Rényi phase transition by Rényi index transition.

---

## Editorial Decision

published